# DECISION TRANSFORMERS AS ZERO-SHOT LEARNERS VIA TEXT-BEHAVIOR ALIGNMENT

## ABSTRACT

Offline meta-reinforcement learning (meta-RL) aims to train agents that can generalize to unseen tasks using pre-collected data from related tasks. Recent approaches leverage the scalability of transformer architectures to model behavior sequences and support task adaptation using target task demonstrations. However, such data is often unavailable in real-world settings, where the task objective may be known but cannot be easily demonstrated. In contrast, humans routinely interpret and perform new tasks based solely on natural language instructions. In this work, we explore the potential of using natural language task descriptions to enable zero-shot task adaptation in offline meta-RL without access to target task data. We propose the Text-Guided Decision Transformer (TG-DT), a framework that enables zero-shot generalization by grounding policy learning in natural language. TG-DT learns a shared embedding space between task descriptions and behavioral trajectories via dual contrastive and matching-based objectives, ensuring robust alignment. A transformer-based policy is then conditioned on these aligned representations to generate task-appropriate actions. At test time, TG-DT synthesizes policies for unseen tasks using only their text descriptions and can optionally leverage a description-guided data sharing strategy to enhance adaptation. Experiments on standard offline meta-RL benchmarks, including MuJoCo and Meta-World, demonstrate that TG-DT achieves strong generalization to unseen tasks.

## 1 INTRODUCTION

Offline reinforcement learning (RL) aims to learn an optimal policy from pre-collected datasets without requiring environment interactions (Levine et al., 2020b). This makes it particularly suitable for real-world applications such as healthcare (Gottesman et al., 2019), robotics (Cabi et al., 2020), and autonomous driving (Kidambi et al., 2020), where collecting online data can be costly or unsafe. A major milestone in this area is the Decision Transformer (DT) (Chen et al., 2021), which reformulates offline RL as a sequence modeling problem and leverages transformers to condition actions on desired returns, enabling the capture of long-horizon dependencies and flexible policy generation.

To extend DT beyond single-task scenarios and enhance its generalization to unseen tasks, DT-based offline meta-RL (Mitchell et al., 2021) has gained increasing attention, which learns across a distribution of tasks to acquire a transferable prior that facilitates adaptation to novel tasks. For instance, Prompt-DT (Xu et al., 2022) introduces expert demonstrations as prompts to encode task-specific information for adaptation. Generalized DT (GDT) (Furuta et al., 2021) utilizes hindsight reward distributions, while Meta-DT (Wang et al., 2024) incorporates a meta-policy to select informative trajectories for adaptation. Although these methods advance generalization to varying degrees, they remain constrained by their reliance on task-specific data (Xu et al., 2022; Wang et al., 2024; Zheng et al., 2024) or the need for access to test environments during adaptation (Xu et al., 2018; Rakelly et al., 2019; Zintgraf et al., 2021; Dong et al., 2024). Such requirements are often impractical in real-world settings, preventing these approaches from achieving true zero-shot generalization and limiting their broader applicability.

**Motivation.** These limitations naturally raise a key research question: *How can agents achieve zero-shot generalization to new tasks without test-time interaction or access to task-specific data?* Although behavioral data from new tasks is often unavailable, high-level task intent is typically known in advance and can be expressed through language. These descriptions convey the goal in

broad, human-like terms (*e.g.*, "open the drawer halfway") without revealing hidden details such as exact coordinates or numeric rewards. For example, a household robot trained on tasks such as "tidying up toys" or "loading the dishwasher" may later be instructed to perform an unseen task, such as "setting the table for dinner", which can be clearly described and semantically related to past experience. This observation motivates our approach: leveraging natural language as a medium for task specification in offline meta-RL. However, enabling this capability presents a unique challenge: aligning language with behavior requires reasoning over temporally extended state-action sequences, varying degrees of task completion, and inconsistent demonstration quality. Effectively modeling this alignment is essential for zero-shot generalization in the absence of task-specific data.

**Our Approach.** In this paper, we propose T̲ext-G̲uided D̲ecision T̲ransformer (TG-DT), a novel offline meta-RL framework that enables zero-shot policy generalization to unseen tasks using only natural language task descriptions without requiring task-specific data or environment interaction at test time. TG-DT aligns behavioral trajectories with text task descriptions by learning a shared representation space through a dual alignment mechanism that combines contrastive learning with a matching-based objective. A DT is also trained to condition its action generation on these text-derived embeddings, effectively grounding task semantics in policy behavior.

At test time, TG-DT achieves zero-shot generalization by leveraging its text-conditioned policy to generate appropriate actions for previously unseen tasks based solely on their natural language descriptions. To further improve adaptation, TG-DT incorporates a description–guided data sharing strategy based on semantic similarity that selectively reuses offline trajectories from semantically related training tasks. This mechanism enhances generalization in the zero-shot setting and naturally extends to few-shot scenarios when limited task-specific data is available. *Our main contributions are as follows:*

- We propose TG-DT, a novel offline meta-RL framework that enables zero-shot generalization to unseen tasks using only text task descriptions. At its core, TG-DT uses a text-conditioned DT, which conditions action inference on text, allowing flexible and generalizable policy execution.
- To bridge the semantic gap between natural language and offline behavioral trajectories, we introduce a dual alignment mechanism that combines contrastive learning and matching-based objectives. This alignment enables robust grounding of textual task descriptions in offline trajectories.
- We enhance TG-DT's adaptability through description–guided data sharing based on semantic similarity, which facilitates effective adaptation by leveraging trajectories from similar tasks.
- Extensive experiments on benchmark offline meta-RL tasks show that TG-DT achieves compatible performance with state-of-the-art baselines in both zero-shot and few-shot generalization. [1]

## 2 PRELIMINARIES

**Offline Meta Reinforcement Learning.** RL is usually formulated as a Markov Decision Process (MDP), defined by $M = \langle \mathcal{S}, \mathcal{A}, Tr, R, \gamma \rangle$, where $\mathcal{S}$ and $\mathcal{A}$ denote the state and action spaces, $Tr$ is the state transition function, $R$ is the reward function, and $\gamma$ is the discount factor. The objective is to learn a policy $\pi(a|s)$ that maximizes the expected return $J_M(\pi) = \mathbb{E}_\pi \left[ \sum_{t=0}^{\infty} \gamma^t R(s_t, a_t) \right]$.

In offline meta-RL, we consider a task distribution $M_i = \langle \mathcal{S}, \mathcal{A}, Tr_i, R_i, \gamma \rangle \sim \mathcal{P}(M)$, where tasks share state and action spaces but differ in dynamics and rewards. For $N$ training tasks $\{M_i\}_{i=1}^N$, the agent receives corresponding offline datasets $\{\mathcal{D}_i\}_{i=1}^N$, where $\mathcal{D}_i = \{(s_{i,j}, a_{i,j}, r_{i,j}, s_{i,j+1})_{j=1}^J\}$ is collected under behavior policy $\pi_\beta^i$. The goal is to learn a meta-policy $\pi_{\text{meta}}$ that generalizes to new tasks. At test time, the agent faces a new task $M_k \sim \mathcal{P}(M)$. In the *few-shot* setting, a small offline data is provided to support adaptation. In the *zero-shot* setting, no additional task-specific data is allowed. We study an even *stricter variant of zero-shot adaptation*, where the agent is not allowed any online interaction with the new task and rely solely on its natural language description. The meta-policy $\pi_{\text{meta}}$ aims to maximize expected performance over test tasks: $J(\pi_{\text{meta}}) = \mathbb{E}_{M \sim P(M)} \left[ J_M(\pi_{\text{meta}}) \right]$.

**Decision Transformer (DT).** DT (Chen et al., 2021) formulates offline reinforcement learning as a return-conditioned sequence modeling problem, inspired by the success of transformers in language modeling (Vaswani et al., 2017). Instead of explicitly modeling MDP dynamics or estimating value functions, DT learns directly from offline behavior data by autoregressively predicting actions. In

---

[1]Our code is available at https://anonymous.4open.science/r/TG-DT-3352/.

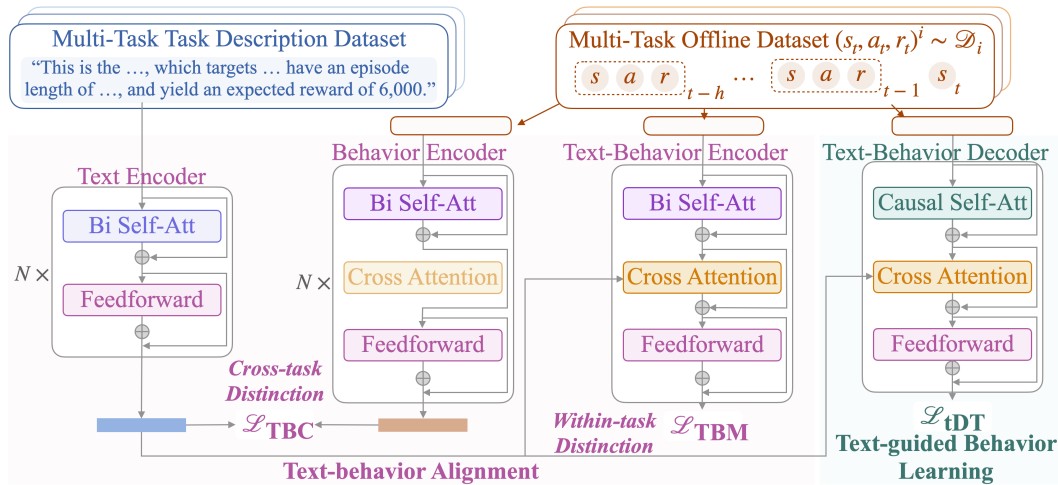

Figure 1: TG-DT model architecture and objectives (same parameters share color). TG-DT employs a text-behavior encoder-decoder framework to enable text-guided behavior learning and alignment: (1) Text and behavior encoders are jointly trained with $\mathcal{L}_{\text{TBC}}$ in Eq. (1) to align representations for cross-task distinction; (2) The text-behavior encoder incorporates cross-attention layers and is further trained with $\mathcal{L}_{\text{TBM}}$ in Eq. (2) to capture within-task differences; (3) Text-behavior decoder replaces bi-directional self-attention with causal self-attention, while reusing the encoder's cross-attention and feedforward layers, and is trained with $\mathcal{L}_{\text{tDT}}$ in Eq. (3) to generate behaviors from task descriptions.

DT, each trajectory is represented as a sequence of tokens $(\hat{R}_t, s_t, a_t)$, where $s_t$ is the state, $a_t$ is the action, and $\hat{R}_t = \sum_{t'=t}^{T} r_{t'}$ denotes the return-to-go (RTG) from timestep $t$. The RTG serves as a conditioning signal, guiding the model to replicate high-return behaviors observed in the dataset. During training, the model is optimized to predict the next action given the sequence of past RTGs, states, and actions. At test time, a target return $G^\star$ is provided, and the RTG is estimated dynamically as $\hat{R}_t = G^\star - \sum_{t'=0}^{t} r_{t'}$. At each timestep, the embeddings of $\hat{R}_t$, $s_t$, and $a_t$ are concatenated and input to the transformer. The model is trained to minimize the prediction error between its output and the ground-truth action.

## 3 TEXT-GUIDED DECISION TRANSFORMER

We present the Text-Guided Decision Transformer (TG-DT), a novel offline meta-RL framework that enables zero-shot adaptation using only natural language task descriptions, as illustrated in Fig. 1.

### 3.1 TASK DESCRIPTIONS IN OFFLINE META-RL

Humans often adapt to novel tasks by leveraging high-level natural language instructions, without direct experience or environmental interaction. These instructions convey task intent based on prior familiarity with the environment and related experiences, enabling people to generalize effectively even in unfamiliar scenarios. Inspired by this capability, we use task descriptions as the sole source of information for zero-shot generalization without online interaction.

To ensure consistency and interpretability, we adopt a templated strategy for constructing task descriptions. For each trajectory $\tau_j \in \mathcal{D}_i$, we create a natural language description in the format: "`This is the [task_name], which targets [task_intent]. Its corresponding environment is [environment_description]. Good demonstrations typically have an episode length of [episode_length], and yield an expected return of [expected_return]. This demonstration yields a [return].`" Each placeholder is deterministically filled using metadata extracted from $\tau_j$, including the task name (*e.g.*, pen-drawer task), high-level intent (without explicit goals, *e.g.*, open a drawer to a specific location), environment information, episode length, and expected return. During training, these values come directly from the offline dataset metadata. This design grounds descriptions in observable behavior while conveying both task intent and environmental context. It

serves as a training signal that grounds environmental information into the text description and also mirrors how humans typically describe tasks, supporting alignment with language-based reasoning.

To reflect the diverse quality of demonstrations, we employ trajectory-level pairing, where each trajectory $\tau_j$ is coupled with its own natural language description $\boldsymbol{p}_j$, rather than using a single description for all trajectories within a task. This pairing captures fine-grained variation in demonstration quality, such as episode length and cumulative reward, by encoding trajectory-specific properties directly into the text.

Formally, our training data consists of a collection of text-behavior pairs $\{(\boldsymbol{p}_j, \tau_j)\}_{j=1}^{|\mathcal{D}_i|}$ where each trajectory $\tau_j$ comes from an offline dataset $\mathcal{D}_i$ for training tasks $M_i \sim \mathcal{P}(M)$. We aim to learn a meta-policy $\pi_{\text{meta}}$ that generalizes to unseen tasks using only their natural language descriptions at test time, thus supporting zero-shot adaptation in the offline setting.

## 3.2 Text-behavior Alignment

A key challenge in text-guided offline meta-RL is enabling it to understand natural language task descriptions. Inspired by recent advances in vision-language alignment (Li et al., 2022; Radford et al., 2021; Eslami & de Melo, 2025), we extend similar strategies to align text and behavior, which introduce unique challenges. Unlike static images, trajectories are temporal sequences that capture dynamic interactions and partial environment information. Text in this context encodes abstract intents, and must be grounded in sequential behavior that unfolds over time. Moreover, offline behavior datasets contain demonstrations of varying quality, requiring the model to develop sensitivity to performance rather than perceptual similarity. These challenges demand a richer alignment mechanism capable of reasoning over temporal dynamics, quality signals, and task semantics.

We introduce a Text-Behavior Alignment (TBA) module that maps task descriptions and behavioral trajectories into a shared latent space. As shown in Fig. 1, TBA consists of three components: a *text encoder*, a *behavior encoder*, and a *text-behavior encoder*. The text and behavior encoders embed descriptions $\boldsymbol{p}_j$ and trajectories $\tau_j$ into $\boldsymbol{e}_j^p$ and $\boldsymbol{e}_j^\tau$, respectively. The text-behavior encoder models their joint interaction. Alignment is guided by complementary objectives: *contrastive learning* captures coarse semantic alignment across tasks, while *matching supervision* enables fine-grained distinction of behavior quality within tasks.

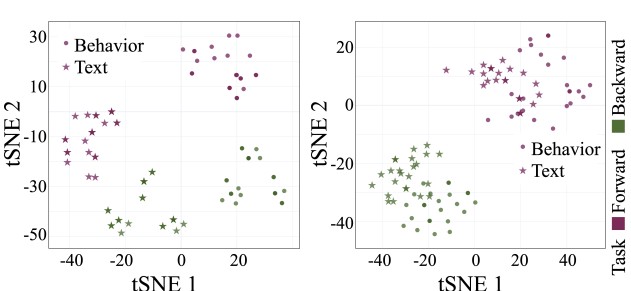

(a) tSNE of TBC-aligned representations.

(b) tSNE of TBC & TBM-aligned representations.

Figure 2: TBM pulls matched text–behavior pairs closer in Cheetah-dir across forward/backward tasks.

**Text-behavior Contrastive Learning (TBC).** To enable cross-task distinction, we introduce a contrastive objective that aligns text and behavior embeddings from the same task while separating those from different tasks. Given a batch of $B$ text-behavior pairs $\{(\tau_j, \boldsymbol{p}_j)\}_{j=1}^B$, and their encoded representations $\{(\boldsymbol{e}_j^\tau, \boldsymbol{e}_j^p)\}_{j=1}^B$, similarity is measured via scaled cosine similarity $\text{sim}(\boldsymbol{e}_j^\tau, \boldsymbol{e}_j^p) = \frac{(\boldsymbol{e}_j^\tau)^\top \boldsymbol{e}_j^p}{\|\boldsymbol{e}_j^\tau\| \|\boldsymbol{e}_j^p\|}$. The *text-behavior contrastive loss* is defined as:

$$\mathcal{L}_{\text{TBC}} \triangleq \frac{1}{B} \sum_{j=1}^B \left[ -\log \frac{\exp\left(\text{sim}(\boldsymbol{e}_j^\tau, \boldsymbol{e}_j^p)/\tau\right)}{\sum_{k=1}^B \exp\left(\text{sim}(\boldsymbol{e}_j^\tau, \boldsymbol{e}_k^p)/\tau\right)} - \log \frac{\exp\left(\text{sim}(\boldsymbol{e}_j^p, \boldsymbol{e}_j^\tau)/\tau\right)}{\sum_{k=1}^B \exp\left(\text{sim}(\boldsymbol{e}_j^p, \boldsymbol{e}_k^\tau)/\tau\right)} \right], \quad (1)$$

where $\tau$ is a temperature hyperparameter controlling the distribution sharpness. The first term aligns each behavior embedding with its corresponding text embedding, while the second does the reverse. Each positive pair is a trajectory and its own description. Negatives are sampled predominantly from different tasks to encourage task-level separation. This design promotes both text-behavior alignment and cross-task distinction, helping the model disambiguate tasks based on semantic intent. By pulling together matched text–behavior pairs and pushing apart those from unrelated tasks, the model acquires a more structured representation of task space which is critical for generalization.

To further enhance consistency, the text and behavior encoders share feedforward layers, promoting alignment in a unified task-centric space (Eslami & de Melo, 2025). We also adopt momentum encoders with soft targets (Li et al., 2021) to improve robustness and reduce false negatives.

**Text-behavior Matching (TBM).** While TBC promotes cross-task distinction, it relies on coarse semantic similarity and struggles to capture fine-grained differences or link language to environment-specific dynamics and data quality variations within the same task. To address this, we introduce a *text-behavior matching* loss, formulated as a *binary classification task* to distinguish matched from mismatched text–behavior pairs. A *text-behavior encoder* processes each text-behavior pair $(\tau_j, \boldsymbol{p}_j)$ and produces a joint embedding, which a linear classification head uses to predict a matching score $z_j$. The loss is then defined as a binary cross-entropy loss:

$$\mathcal{L}_{\text{TBM}} \triangleq -\frac{1}{B} \sum_{j=1}^{B} \left[ y_j \log \sigma(z_j) + (1 - y_j) \log(1 - \sigma(z_j)) \right], \tag{2}$$

where $y_j \in \{0, 1\}$ indicates whether the text-behavior pair is a true match, and $\sigma(\cdot)$ is the sigmoid function. The text-behavior encoder features cross attention layers to directly model interactions between text and behaviors, embedding environment-specific signals such as rewards and transitions. To improve the differentiation on data quality, we adopt the hard negative mining strategy (Li et al., 2021): negatives include both inter-task mismatches and intra-task mismatches (*e.g.*, a text paired with a trajectory of different quality). This ensures the model to discriminate subtle differences in return or success, not just coarse task identity. As shown in Fig. 2b, TBM enhances within-task alignment by bringing matched pairs closer in the shared space.

### 3.3 Text-guided Behavior Learning

We enable policy learning from task descriptions via text-guided behavior learning. TG-DT extends Decision Transformer (DT) to a *text-behavior decoder* that autoregressively generates actions based on past trajectories and the task descriptions. It is optimized with the *text-conditioned DT* loss:

$$\mathcal{L}_{\text{tDT}} \triangleq \mathbb{E}_{M_i \sim P(M), \tau_j \sim \mathcal{D}_i} \left[ \sum_{t=1}^{T} \| \boldsymbol{a}_{j,t} - \pi(\tau_{j,t-1}, \boldsymbol{p}_j) \|^2 \right]. \tag{3}$$

Here, $\pi(\cdot)$ is the policy conditioned on prior trajectory $\tau_{j,t-1}$ and task description $\boldsymbol{p}_j$. The mean squared error (MSE) loss trains the policy to reproduce expert actions consistent with the task description, extending DT beyond return-to-go conditioning to language-aligned embeddings for zero-shot task specification.

During meta-training, we jointly optimize three objectives, *i.e.*, $\mathcal{L} = \mathcal{L}_{\text{TBC}} + \mathcal{L}_{\text{TBM}} + \mathcal{L}_{\text{tDT}}$, enabling the model to separate tasks, capture trajectory-level quality differences, and generate actions consistent with task intent. This combined loss couples alignment with policy learning, enabling TG-DT to adapt to behavior data while remaining grounded in text and thus bridging the gap between language and behavior in offline meta-RL.

To strengthen language understanding, we initialize TG-DT components using the pre-trained BLIP model (Li et al., 2022) due to its capability to capture the rich text semantics. Although BLIP was originally designed for image–text tasks, its encoders learn cross-modal attention patterns that transfer effectively to trajectory–text alignment.

## 4 Description-Guided Task Adaptation

**Zero-Shot Adaptation from Text Descriptions.** Given a new task $M_k$ defined solely by its natural language description $\boldsymbol{p}_k$, TG-DT enables zero-shot adaptation by conditioning the policy on the aligned text embedding. At test time, $\boldsymbol{p}_k$ follows the same templated format as training prompts, but with values such as expected return and episode length replaced by approximate statistics inferred from the training distribution rather than using ground-truth values[2]. During inference, $\boldsymbol{p}_k$ is passed to the text encoder to get $\boldsymbol{e}_k^p$, which guides the text-behavior decoder to generate actions autoregressively.

---

[2]See Appx. E for test prompts.

Since the model is meta-trained to align text and behavior in a shared latent space, $e_k^p$ effectively captures the task semantics and supports generalization to unseen tasks. *This process requires no environment interaction or task-specific data at test time.* Instead, generalization emerges from the model's ability to associate semantic features in natural language with corresponding behavioral patterns observed during training. This makes TG-DT well-suited for applications where test-time data collection is infeasible, costly, or unsafe.

**Description-Guided Data Sharing for Adaptation Enhancement.** To further enhance zero-shot performance, we incorporate a data-sharing strategy based on semantic similarity. Given a test task description $p_k$ with embedding $e_k^p$, we retrieve the top-$K$ training task descriptions whose text embeddings are most similar to $e_k^p$ via $\arg\max_{p_j \in \cup_{i=1}^N \{p_j\}_{j=1}^{|\mathcal{D}_i|}} \text{sim}(e_k^p, e_j^p)$. We then use the corresponding trajectories to fine-tune the text-behavior decoder with $\mathcal{L}_{tDT}$ in Eq. (3), while still conditioning on $p_k$. This allows the model to refine its behavior using related training data without requiring any supervision from the target task.

**Few-Shot Adaptation from Text Descriptions.** When a small amount of task-specific offline data $\mathcal{D}_k$ is available, the text-behavior decoder can be fine-tuned in a few-shot setting. It leverages both the task description $p_k$ and limited demonstrations to improve adaptation. We use the same $\mathcal{L}_{tDT}$ objective in Eq. (3), treating $\mathcal{D}_k$ as a lightweight fine-tuning set. This hybrid approach retains the generalization benefits of language guidance while allowing for task-specific refinement.

## 5 EXPERIMENT

We evaluate TG-DT's generalization capability on standard benchmarks to answer the following key questions: *1).* Can TG-DT outperform strong baselines in zero- and few-shot generalization to unseen tasks? (See Sec. 5.2) *2).* Does the text-behavior alignment mechanism produce a meaningful shared embedding space that supports task understanding and transfer? (See Sec. 5.3) *3).* What is the impact of the contrastive (TBC) and matching (TBM) components on generalization performance, and how effective is the description-guided data sharing strategy in enhancing adaptation to new tasks? (See Sec. 5.4) *4).* How robust is TG-DT to variations in offline data quality? (See Sec. 5.5)

### 5.1 EXPERIMENT SETUP

**Environments.** We evaluate TG-DT on two widely adopted benchmarks, MuJoCo (Todorov et al., 2012) and MetaWorld (Yu et al., 2020). For MuJoCo, we follow Xu et al. (2022) and evaluate on Cheetah-dir, Cheetah-vel, and Ant-dir. For MetaWorld, we use the ML10, and ML45 task suites for robotic manipulation. Datasets are collected using SAC (Haarnoja et al., 2018) trained independently on sampled tasks, with three dataset types: Medium, Mixed, and Expert.

**Baselines.** We compare TG-DT with two categories of baselines. *DT-based baselines* include: *1).* Prompt-DT (PDT) (Xu et al., 2022): Builds on DT by introducing trajectory-level prompts and reward-to-go conditioning to enable generalization to unseen tasks; *2).* Generalized DT (GDT) (Furuta et al., 2021): Extends DT by leveraging hindsight reward distributions to guide adaptation and improve generalization; *3).* Meta-DT (MDT) (Wang et al., 2024): Enhances DT by introducing a meta-policy to enable stronger generalization;*4).* Hyper DT (HDT) (Xu et al., 2023): Adapts DT to new tasks via a lightweight adaptation module, allowing efficient fine-tuning with minimal data; and *5).* DPDT (Zheng et al., 2024): Augments DT with decomposed prompt tuning, enabling parameter-efficient test-time adaptation. *Language-conditioned RL baselines* include *6).* BC-Z (Jang et al., 2022) achieves zero-shot generalization from language via imitation learning, and *7).* BAKU (Haldar et al., 2024) introduces an efficient transformer for multi-task policy learning. We delegate more environment descriptions, hyperparameters and implementation details in Appx. C.

### 5.2 ZERO-SHOT AND FEW-SHOT GENERALIZATION

**Zero-shot Generalization.** Results in Tab. 1 show that TG-DT performs on par with or better than DT-based baselines. Unlike DT-based baselines (*e.g.*, Meta-DT, and HDT), which assume access to test task interaction, TG-DT generalizes strictly from text intent without any environment interaction. Language-conditioned baselines such as BC-Z and BAKU underperform in this setting, as they rely on imitation without return conditioning. TG-DT's dual alignment mechanism leverages task intent in

| Method | Cheetah-dir | Cheetah-vel | Ant-dir | ML10 | ML45 |
|---|---|---|---|---|---|
| PDT† | 548.9 | -150.6 | 214.2 | 289.2 | 248.3 |
| GDT | 129.2 | -218.4 | 167.9 | 169.8 | 153.2 |
| MDT† | 539.6 | -102.7 | **357.5** | 335.2 | 306.4 |
| HDT† | 445.3 | -162.7 | 215.4 | 266.4 | 245.7 |
| DPDT† | 548.1 | -142.6 | 321.8 | 360.4 | **311.8** |
| BC-Z† | 310.5 | -123.9 | 254.9 | 199.7 | 157.4 |
| BAKU | 348.6 | -110.5 | 277.2 | 216.3 | 163.2 |
| TG-DT | **549.9** | **-93.0** | 328.3 | **361.1** | 309.6 |

| Method | Cheetah-dir | Cheetah-vel | Ant-dir | ML10 | ML45 |
|---|---|---|---|---|---|
| PDT† | 587.3 | -136.8 | 310.8 | 322.2 | 298.8 |
| GDT | 569.7 | -118.9 | 291.3 | 211.6 | 209.3 |
| MDT† | 599.4 | -95.3 | 409.9 | 361.3 | 442.5 |
| HDT† | 489.5 | -122.5 | 376.7 | 319.3 | 315.4 |
| DPDT† | 591.4 | -78.3 | 410.4 | **425.6** | 498.6 |
| BC-Z† | 356.2 | -126.5 | 297.4 | 217.4 | 163.9 |
| BAKU | 323.8 | -142.8 | 356.2 | 309.2 | 172.8 |
| TG-DT | 598.4 | **-73.2** | **412.9** | 421.5 | **499.2** |

Table 1: Zero-shot test returns of TG-DT *vs.* baselines using Medium datasets. We report average returns over 5 runs (higher is better), with standard deviations in Appx. C. † denotes methods that require test-time interaction to obtain adaptation demonstrations. TG-DT requires no interaction.

Table 2: Few-shot test returns of TG-DT *vs.* baselines using Medium datasets. We report average returns over 5 runs (higher is better), with standard deviations in Appx. C. † denotes methods that require test-time interaction to obtain adaptation demonstrations. TG-DT requires no interaction.

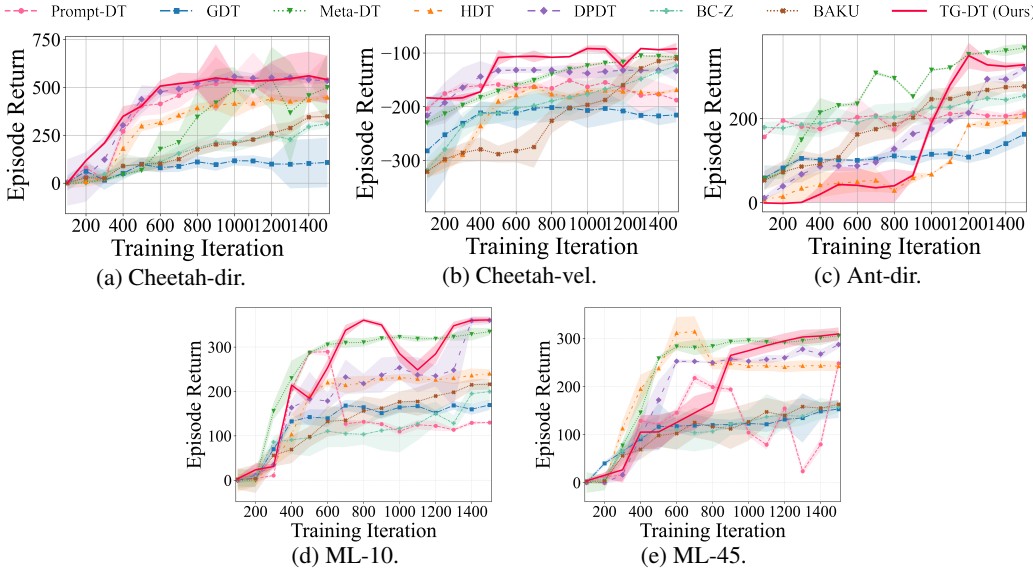

Figure 3: The received return curves averaged over test tasks of TG-DT and baselines using Medium datasets in the zero-shot setting.

text while grounding it in trajectory quality, enabling robust generalization across unseen tasks. Fig. 3 further illustrates TG-DT's stable convergence and stable performance during training. The slower convergence in ML10 and ML45 tasks shows a trade-off resulting from TG-DT's deliberate emphasis on learning robust, task-conditioned representations rather than rapidly adapting to individual tasks. Overall, the results demonstrate TG-DT's ability to learn robust, generalizable policies from task descriptions without sacrificing performance in zero-data settings.

**Few-shot Generalization.** As shown in Tab. 2, TG-DT demonstrates strong performance under the few-shot setting, consistently outperforming or matching all baselines across most environments. This improvement can be attributed to TG-DT's ability to effectively leverage real demonstrations from previously unseen tasks at test time. By conditioning on these demonstrations, TG-DT refines its task understanding and adapts its behavior more precisely to the new task.

### 5.3 TEXT-BEHAVIOR ALIGNMENT PERFORMANCE

To evaluate alignment between task descriptions and behavior trajectories, we visualize their embeddings using tSNE and assess their similarity via cosine similarity. Fig. 4 shows results for Cheetah-vel

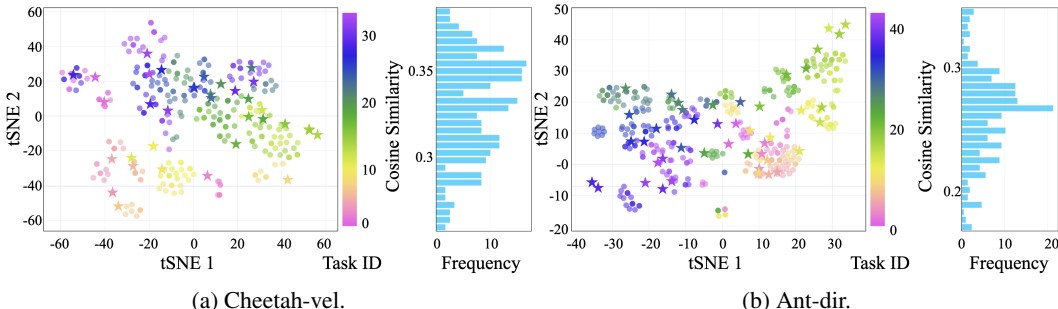

(a) Cheetah-vel.

(b) Ant-dir.

Figure 4: t-SNE and cosine similarity histograms of text–behavior representations. Left: t-SNE with text centroids (⋆) and behavior embeddings (●); Right: cosine similarity distributions.

and Ant-dir. In both environments, the text embeddings (⋆) and corresponding behavior embeddings (●) form coherent clusters in the projected space, suggesting that the model learns a semantically meaningful alignment between language and behavior. For Cheetah-vel (Fig. 4a), the embeddings are organized along a smooth manifold with a clear progression across different task IDs. This reflects the continuous nature of the velocity control tasks, which are sampled from a uniform distribution over a range of target velocities. In Ant-dir (Fig. 4b), a similarly structured embedding space emerges, where tasks with similar directional goals are grouped together, despite the more complex and higher-dimensional action space associated with limb coordination in the Ant environment.

The adjacent histograms display the distribution of cosine similarity scores between each text-behavior pair. For Cheetah-vel, the similarity values are more concentrated around higher values (mean ≈ 0.34), indicating a tighter alignment. In contrast, Ant-dir shows a broader spread and slightly lower mean similarity (≈ 0.28), likely due to the greater task diversity and motor complexity. Despite this, the overall average cosine similarity remains close to 0.3 in both cases. When coupled with the performance metrics in Tab. 1, these findings suggest that the representation gap between text and behavior does not hinder policy learning. This observation aligns with recent work on multi-modal representation learning (Jiang et al., 2023), which shows that a moderate discrepancy in embedding spaces can be tolerated as long as the overall structure is preserved and contrastive alignment is effective.

## 5.4 ABLATION STUDY

**Impact of TBM and TBC.** Tab. 3 shows the performance of TG-DT when selectively disabling Text-Behavior Contrastive Learning (TBC) and Text-Behavior Matching (TBM) during training. Removing both TBC and TBM significantly degrades performance across all environments. When TBC is ablated, we observe unstable task adaptation and lower returns, suggesting that the model struggles to enforce behavioral coherence between different task instances. Besides, removing TBM leads to a less precise alignment between the task prompt and behavior trajectories, resulting in reduced task relevance. In contrast, TG-DT with both parts active consistently achieves the highest performance, indicating that these two components play complementary roles in encouraging effective model generalization and text-behavior alignment.

**Effect of the Shared Demonstration Count K.** Fig. 5 illustrates the effect of varying the number of shared demonstrations used during description-guided data sharing. No data-sharing is implemented when $K = 0$. We observe that incorporating a small number of real trajectories from related tasks (*i.e.*, $K = 1$ or $K = 2$) boosts performance in

| Environment | w/o TBC, TBM | w/o TBC | w/o TBM | TG-DT |
|---|---|---|---|---|
| Cheetah-dir | 859.4 | 936.8 | 875.2 | **958.4** |
| Cheetah-vel | −55.7 | −24.6 | −46.6 | **−21.4** |
| Ant-dir | 133.6 | 322.1 | 298.7 | **383.4** |

Table 3: The impact of TBC and TBM on TG-DT.

settings such as Ant-dir and ML45. This indicates even minimal shared data can provide valuable contextual grounding for TG-DT's task representations. However, performance gains saturate or slightly decline as $K$ increases beyond a certain point, suggesting diminishing returns and potential overfitting to related tasks instead of the test task.

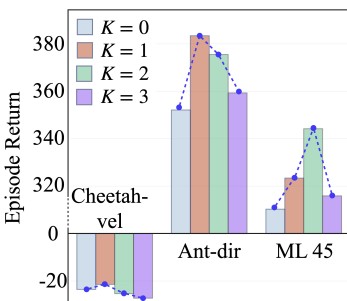

Figure 5: Impact of data sharing fine-tuning with varying numbers of shared demonstrations.

Table 4: Zero-shot test returns of TG-DT against baselines using Mixed and Expert datasets. We report the average returns and standard deviations for five runs (the higher, the better).

| **Expert** | PDT | GDT | MDT | HDT | DPDT | BC-Z | BAKU | TG-DT |
|---|---|---|---|---|---|---|---|---|
| Cheetah-dir | 929.7 | 779.3 | 947.2 | 875.2 | 955.1 | 759.2 | 761.3 | **958.4** |
| Cheetah-vel | −39.3 | −59.2 | −29.7 | −45.3 | −30.1 | −61.4 | −55.3 | **−21.4** |
| Ant-dir | 347.1 | 352.2 | 376.4 | 361.49 | **384.2** | 309.5 | 337.6 | 383.4 |

| **Mixed** | PDT | GDT | MDT | HDT | DPDT | BC-Z | BAKU | TG-DT |
|---|---|---|---|---|---|---|---|---|
| Cheetah-dir | 836.6 | 699.1 | 853.6 | 786.0 | **861.2** | 687.4 | 693.6 | 830.1 |
| Cheetah-vel | −44.3 | −64.2 | −34.7 | −50.3 | −35.1 | −72.3 | −63.3 | **−34.4** |
| Ant-dir | 310.4 | 307.6 | 327.0 | 314.6 | 342.6 | 291.4 | 301.3 | **344.9** |

### 5.5 ROBUSTNESS TO THE QUALITY OF OFFLINE DATASETS

To evaluate TG-DT's robustness to varying offline training data qualities, we conduct experiments using Expert, Mixed, and Medium datasets and compare zero-shot performance against baselines. As shown in Tab. 1 and Tab. 4, TG-DT consistently delivers strong performance across all dataset types.

On Expert datasets, TG-DT achieves top-tier performance, matching or surpassing existing baselines across most tasks. This is expected, as expert trajectories offer consistent, high-reward behavior that aligns well with TG-DT's task-conditioned modeling approach. In contrast, Mixed datasets present a more challenging setting due to the inclusion of noisy or suboptimal data. While all methods experience performance drop, TG-DT remains competitive and even outperforms alternatives like DPDT and Meta-DT in complex environments such as Ant-dir. This resilience highlights TG-DT's ability to filter out task-irrelevant patterns through its semantic task-behavior alignment.

## 6 RELATED WORK

**Offline RL** learns from fixed data without environment interactions (Levine et al., 2020b; Konyushkova et al., 2020), and extensions to meta-RL improve generalization but typically require test-time demonstrations or adaptation data (Xu et al., 2022; Wang et al., 2024; Zheng et al., 2024; Xu et al., 2018; Rakelly et al., 2019; Zintgraf et al., 2021). **Language-conditioned RL** offers another perspective: BC-Z (Jang et al., 2022) and BAKU (Haldar et al., 2024) omit reward conditioning and focus on imitation, TTCT (Dong et al., 2024) assumes online environment access, and a concurrent effort (Zhang et al., 2025) prepends text to trajectories but performs only implicit alignment and overlooks trajectory quality. TG-DT instead introduces explicit dual alignment to capture both inter-task semantics and intra-task variation, ensuring robustness to noisy offline data. Finally, **multi-modal alignment methods** such as CLIP (Radford et al., 2021) and BLIP (Li et al., 2022) show the power of joint embedding spaces but target static perception, while TG-DT extends alignment to sequential decision-making by grounding temporally extended trajectories in language under return conditioning. (See Appx. B for an extended discussion.)

## 7 CONCLUSIONS & LIMITATIONS

We proposed TG-DT, a Text-Guided Decision Transformer that leverages text task descriptions and text-behavior alignment objectives to improve generalization in offline meta-RL. Extensive experiments across diverse tasks and dataset qualities show that TG-DT consistently outperforms or matches strong baselines in both zero-shot and few-shot settings. Our ablation studies confirm the effectiveness of the proposed TBC and TBM modules, as well as the benefit of limited shared data. TG-DT remains robust under suboptimal data conditions, making it a practical and data-efficient solution for real-world offline meta-RL.

**Limitation.** TG-DT relies on templated task descriptions that include metadata during training, and at test time it replaces these fields with approximate values inferred from the training tasks. This design prevents oracle information leakage but reduces robustness to free-form natural language, making support for unconstrained instructions an important direction for future work.

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
