# 8 APPENDIX

## A ETHICS AND REPRODUCIBILITY STATEMENTS

**Ethics Statement** This research adheres to the ICLR Code of Ethics. All experiments are conducted on publicly available benchmark datasets and environments that do not involve human subjects or personally identifiable information. No new data collection was performed, and no sensitive or private information is included. The contributions of this work are methodological, aiming to advance machine learning. While reinforcement learning and related methods may be applied in safety-critical or socially sensitive domains, this paper does not directly address or deploy in such contexts. We have taken care to report results honestly, acknowledge limitations, and follow best practices for research integrity. No conflicts of interest or ethical concerns beyond standard research conduct arise from this work.

**Reproducibility Statement** We have made every effort to ensure the reproducibility of our results. Detailed descriptions of the proposed algorithms, theoretical assumptions, and derivations are provided in the main text and appendices. Hyperparameter settings, model architectures, and training configurations are reported in full. Data preprocessing procedures and evaluation protocols are documented, and all datasets used are publicly available. An anonymous link to source code and instructions for reproducing experiments is included in the paper. Together, these resources ensure that independent researchers can reproduce and verify our findings.

## B EXTENDED RELATED WORK

**Policy Learning as Sequence Modeling.** Decision Transformer (DT) (Chen et al., 2021) reframes RL as sequence modeling, using transformers to predict actions from past states and target returns. This framework has inspired extensions in unsupervised pretraining (Schmied et al., 2023), trajectory ranking (Lee et al., 2022), and multi-task learning (Reed et al., 2022). Prompt-based variants, such as Prompt-DT (Xu et al., 2022) and DPDT (Zheng et al., 2024), adapt to new tasks using soft or handcrafted prompts but lack semantic interpretability and rely heavily on trajectory-derived inputs. GDT (Furuta et al., 2021) uses hindsight rewards for adaptation, while Meta-DT (Wang et al., 2024) employs a meta-policy to select informative trajectories. In contrast, our approach leverages natural language task descriptions to align text and behavior, enabling zero-shot generalization in offline meta-RL. While inspired by recent efforts linking language and control (Brown et al., 2020; Brandfonbrener et al., 2022), we explicitly ground task semantics in behavioral data via task-guided sequence modeling.

**Offline Meta-RL.** Offline RL aims to learn policies from fixed datasets without further interaction with the environment (Levine et al., 2020b; Konyushkova et al., 2020). Offline meta-RL extends this to task distributions, enabling generalization to unseen tasks using only offline data (Mitchell et al., 2020; Pong et al., 2021; Dorfman et al., 2021; Li et al., 2020). Prior methods often rely on temporal-difference (TD) learning, either with optimization-based meta-learning (Finn et al., 2017; Mitchell et al., 2020; Xu et al., 2018) or context-based inference (Yuan & Lu, 2022; Rakelly et al., 2019; Zintgraf et al., 2021; Li et al., 2024). Context-based approaches, such as PEARL (Rakelly et al., 2019) and VariBAD (Zintgraf et al., 2021), learn latent task representations from context trajectories to condition the policy. However, TD-based algorithms are prone to instability due to the deadly triad—bootstrapping, function approximation, and off-policy data (Levine et al., 2020a). Many also depend on hand-crafted constraints to keep policies within the support of the offline data distribution (Ajay et al., 2022). These challenges motivate alternative approaches that avoid bootstrapping and allow more stable, flexible generalization across tasks.

**Language-conditioned RL.** Language offers a natural way to specify tasks, and several methods explore this space. BC-Z (Jang et al., 2022) and BAKU (Haldar et al., 2024) demonstrate language-conditioned imitation learning, but both omit reward conditioning and are not designed for offline meta-RL. TTCT (Dong et al., 2024) leverages text for constraint decomposition but requires online environment interaction to adapt, limiting its applicability in offline domains. MineCLIP (Fan et al., 2022) and STEVE-1 (Lifshitz et al., 2023) show that large-scale vision-language pretraining

can ground behavior in text, but they operate in online or imitation learning settings and are not return-conditioned. Most recently, a concurrent work T2DA Zhang et al. (2025) introduces prompt-augmented trajectories by prepending natural language to behavior sequences. While promising, this approach performs implicit alignment between text and trajectories and does not address intra-task variation or the robustness issues posed by noisy offline data. By comparison, TG-DT introduces explicit dual alignment losses: contrastive learning ensures inter-task separation, while matching-based supervision captures intra-task quality differences. This design directly addresses the variability of offline data and leads to more reliable text-to-behavior grounding.

**Vision-Language Alignment.**    Multi-modal models such as CLIP (Radford et al., 2021) and BLIP (Li et al., 2022) highlight the effectiveness of aligning text and images through large-scale contrastive learning. However, these approaches are designed for static perception tasks. Extending their ideas to sequential decision-making presents unique challenges: trajectories unfold over time, rewards vary in quality, and policies must generate coherent action sequences. TG-DT bridges this gap by aligning natural language with temporally extended trajectories, grounding sequential decision-making in language while preserving return conditioning. This shift from static alignment to dynamic control is central to TG-DT's novelty.

**Summary.**    In summary, prior offline meta-RL methods rely on test-time demonstrations or interaction, while existing language-conditioned approaches either assume online adaptation or lack reward conditioning. TG-DT uniquely combines (1) a stricter offline assumption—zero-shot generalization from language only, with no test-time data; (2) a dual alignment mechanism that robustly links text to behavior across and within tasks; and (3) return-conditioned decision modeling to preserve the strengths of DT while extending it to language-based task specification. This combination positions TG-DT as a novel framework to achieve practical, zero-shot offline generalization guided solely by natural language.

## C  IMPLEMENTATION DETAILS & ENVIRONMENT

**Implementation Details.**    All results are averaged over 5 random seeds with standard deviation shown. Experiments were conducted on 4 NVIDIA RTX 6000 Ada GPUs (48GB), using PyTorch and the Hugging Face Transformers library (Wolf et al., 2019).

We evaluate **TG-DT** on two widely used benchmarks—**MuJoCo** (Todorov et al., 2012) and **Meta-World** (Yu et al., 2020)—to ensure fair and rigorous comparisons. These benchmarks pose diverse challenges such as sparse rewards, high-dimensional continuous control, and complex task variations, making them well-suited for testing generalization in sequence-based offline RL.

**MuJoCo Tasks.**    We follow the experimental protocol from Prompt-DT (Xu et al., 2022) and evaluate TG-DT on three locomotion tasks: **Cheetah-dir**, **Cheetah-vel**, and **Ant-dir**, where agents are penalized for high-magnitude control signals. These environments are designed to assess both zero-shot and few-shot generalization across varied task specifications.

- **Cheetah-dir**: Consists of two tasks—running forward and backward. The agent receives a reward proportional to its velocity along the goal direction. Both training and test sets contain these two tasks.
- **Cheetah-vel**: Contains 40 tasks, each with a target velocity uniformly sampled from $[0, 3]$. The agent is penalized based on the $l_2$ distance from the target velocity. We hold out 5 tasks for testing and train on the remaining 35.
- **Ant-dir**: Includes 50 tasks with goal directions uniformly sampled in 2D space. The 8-joint ant is rewarded for velocity along the specified direction. We use 5 tasks for testing and 45 for training.

**Meta-World Tasks.**    We use the **ML10** and **ML45** task suites, which involve controlling a Sawyer robot's end-effector to reach task-specific goals in 3D space. These benchmarks are standard in meta-RL research and provide a diverse set of robotic manipulation tasks.

- **ML10**: Comprises 10 training and 3 test tasks. Each task requires the robot to move its end-effector to a unique target location. The agent directly controls the XYZ position of the end-effector.

Table 5: Training and testing task indexes when testing the generalization ability in unseen tasks.

| | |
|---|---|
| **Cheetah-dir** | |
| Training set of size 2 | [0, 1] |
| Testing set of size 2 | [0, 1] |
| **Cheetah-vel** | |
| Training set of size 35 | [0–1, 3–6, 8–14, 16–22, 24–25, 27–39] |
| Testing set of size 5 | [2, 7, 15, 23, 26] |
| **Ant-dir** | |
| Training set of size 45 | [0–5, 7–16, 18–22, 24–29, 31–40, 42–49] |
| Testing set of size 5 | [6, 17, 23, 30, 41] |
| **Meta-World ML10** | |
| Training set of size 10 | [0, 9, 19, 29, 33, 36, 39, 40, 48, 49] |
| Testing set of size 3 | [11, 24, 41] |
| **Meta-World ML45** | |
| Training set of size 45 | [0–10, 12–16, 18–24, 26–35, 37–40, 42–49] |
| Testing set of size 5 | [11, 17, 25, 36, 41] |

- **ML45**: Contains 45 training and 5 unseen test tasks, each with a different manipulation goal involving position or object interaction.

**Dataset Construction.**   For MuJoCo tasks, we follow the offline data generation procedure used in (Mitchell et al., 2021), collecting trajectories using SAC (Haarnoja et al., 2018) or TD3 (Fujimoto et al., 2018), and applying penalties on large control inputs. We consider three dataset types:

- **Expert**: Generated using a well-trained SAC/TD3 policy.
- **Mixed**: Combines trajectories from partially and fully trained policies.
- **Medium**: Collected from mid-performance policies.

For Meta-World tasks, we use expert demonstrations generated by scripted controllers provided in the environment suite.

**Task Splits.**   Following Prompt-DT (Xu et al., 2022), we illustrate the training and testing task distributions for each benchmark in Table 5. All experiments strictly follow these splits for consistency.

**BAKU (Haldar et al., 2024) with Few-Shot Extension.**   BAKU is a multi-task imitation learning approach. For fair comparison, we evaluate a variant of BAKU where the few-shot dataset from the target task is included as additional training data. This effectively treats the unseen task as an extra training task, which is outside BAKU's original design (it assumes multi-task training without task-specific adaptation). While this extension provides BAKU with privileged access to the few-shot trajectories, it still underperforms TG-DT. This highlights that simply exposing BAKU to more data is insufficient; TG-DT's advantages stem from its return conditioning and explicit text–behavior alignment, which enable robust generalization under strict offline meta-RL constraints.

**Complete Results.**   Tab. 6 include the complete results of zero-shot setting.

**Implementation Details.** All evaluated methods are carried out with 5 different random seeds, and the mean of the received return is plotted with standard deviation. All experiments were conducted on a server equipped with 4 NVIDIA RTX 6000 Ada GPUs (48GB each), using PyTorch and the Hugging Face Transformers library (Wolf et al., 2019).

## D   HYPERPARAMETERS CONFIGURATION

We show the hyperparameter of TG-DT in Table 7.

| Method | Cheetah-dir | Cheetah-vel | Ant-dir | ML10 | ML45 |
|---|---|---|---|---|---|
| Prompt-DT[†] | 548.9±185.2 | -150.6±12.7 | 214.2±6.2 | 289.2±10.5 | 248.3±11.3 |
| GDT | 129.2±106.5 | -218.4±15.8 | 167.9±18.5 | 169.8±11.2 | 153.2±17.9 |
| Meta-DT[†] | 539.6±14.7 | -102.7±3.4 | **357.5**±9.2 | 335.2±8.7 | 306.4±10.7 |
| HDT[†] | 445.3±13.2 | -162.7±20.5 | 215.4±10.2 | 266.4±8.8 | 245.7±12.3 |
| DPDT[†] | 548.1±12.1 | -142.6±17.9 | 321.8±8.0 | 360.4±6.2 | **311.8**±9.1 |
| BC-Z[†] | 310.5±23.0 | -123.9±10.1 | 254.9±10.1 | 199.7±21.2 | 157.4±10.9 |
| BAKU | 348.6±36.8 | -110.5±7.9 | 277.2±12.6 | 216.3±12.0 | 163.2±9.7 |
| TG-DT (Ours) | **549.9**±176.2 | **-93.0**±10.2 | 328.3±5.0 | **361.1**±5.1 | 309.6±13.4 |

Table 6: Zero-shot test returns of TG-DT against baselines using Medium datasets. We report average returns $\pm$ standard deviations over five runs (higher is better). † denotes methods that require test-time environment interaction to obtain adaptation demonstrations. TG-DT requires no interaction.

Table 7: Hyperparameters used in our experiments.

| Hyperparameters | Value |
|---|---|
| Pretraining model | BLIP |
| $K$ (number of shared data) | 1 |
| Training batch size | 16 |
| Number of evaluation episodes | 50 |
| Learning rate | 1e-4 |
| Weight decay | 1e-4 |
| Number of layers | 3 |
| Number of attention heads | 1 |
| Embedding dimension | 128 |
| Activation | ReLU |
| $r$ | 1 |
| Dropout | 0.1 |
| Device | CUDA |
| Max iterations | 5000 |
| Warmup steps | 10000 |
| Save interval | 500 |

# E   EXAMPLE TASK DESCRIPTIONS

Below we list the example task descriptions of the five testing tasks, where task statistics and intents are inferred and estimated from training tasks:

- **Cheetah-dir**: "This is the Cheetah-dir task, which targets running in the forward direction. Its corresponding environment is the MuJoCo HalfCheetah-v2, a planar 9-DoF robot trained to maximize velocity along the X-axis. Good demonstrations typically have an episode length of 1000 steps and yield an expected return of 1,000. This demonstration yields a return of 1,000."
- **Cheetah-vel**: "This is the Cheetah-vel task, which targets maintaining a velocity of around 2.3 m/s. Its corresponding environment is the MuJoCo HalfCheetah-v2, where the agent is penalized by the squared distance from the target velocity at each timestep. Good demonstrations typically have an episode length of 1000 steps and yield an expected return of -20. This demonstration yields a return of -20."
- **Ant-dir**: "This is the Ant-dir task, which targets movement toward around 135 degrees. Its corresponding environment is an 8-joint quadruped ant robot. Good demonstrations typically have an episode length of 1000 steps and yield an expected return of 1000. This demonstration yields a return of 1000."
- **MetaWorld ML10**: "This is the ML10 reach-target task, which targets moving the end-effector to a designated 3D location. Its corresponding environment is a tabletop robotic arm setup using the Sawyer robot. Good demonstrations typically have an episode length of 150 steps and yield an expected return of 550. This demonstration yields a return of 550."
- **MetaWorld ML45**: "This is the ML45 open-drawer task, which targets opening a drawer to a specific position. Its corresponding environment is a robotic manipulation scene involving multiple

object interactions. Good demonstrations typically have an episode length of 200 steps and yield an expected return of 450. This demonstration yields a return of 450."