# OpenReview forum: "Decision Transformers As Zero-Shot Learners via Text-Behavior Alignment"
_ICLR.cc/2026/Conference — Submitted to ICLR 2026_

### Official Review · Reviewer_ex7f · 2025-10-24

**Soundness:** 2
**Presentation:** 2
**Contribution:** 1
**Rating:** 2
**Confidence:** 4

**Summary:**

This paper proposes a way to train a decision transformer with textual descriptions. The problem setting is as follows: we are given trajectories with text labels, and we want to train a multi-task policy that can handle unseen task descriptions at test time, mainly in a zero-shot manner. To do this, the authors first encode text descriptions into a latent space using contrastive learning with behaviors. With the learned text embeddings, they train a decision transformer on the dataset trajectories. This method is called the text-guided decision transformer (TG-DT). The authors evaluate TG-DT on several HalfCheetah, Ant, and Meta-World tasks, comparing its performance with previous DT and language-conditioned RL methods.

**Strengths:**

* The problem setting is important and the proposed method is reasonable.
* The paper provides several ablation studies about each component of the method, which helps understand their relative importance.

**Weaknesses:**

* I think the biggest weakness of this paper is its empirical evaluation. The authors only evaluate their method on "classic" tasks like HalfCheetah, Ant, and Meta-World. They are not the most "natural" language-based tasks either, and it is unclear how this method works on *actual* language-based benchmarks, such as CALVIN, LIBERO, or the one used in UniPi. Evaluating on classic tasks would be fine if the paper were more theory-oriented, but since this paper is purely empirically motivated and based on scalable transformer learning, I believe it is important to demonstrate its ability on tasks that the community currently finds more relevant.
* Moreover, even on these "classic" tasks, their performance gain seems marginal (Tables 1 and 2). This makes the additional complexity introduced in this paper questionable. I suspect this might again stem from the simplicity of the evaluation tasks.
* I'm not sure the term "offline meta-RL" is appropriate. To me, the problem setting considered in this paper is simply language-conditioned RL (and/or multi-task RL), with a focus on generalization toward unseen tasks. Meta-RL typically involves additional online rollouts or task-specific offline datasets for quick *adaptation*, while this paper mainly focuses on the zero-shot setting (I know the paper also has few-shot adaptation experiments, but they are based on naive fine-tuning with no specific meta-RL techniques, unlike typical meta-RL methods like MAML, PEARL, etc.). Are there similar previous offline RL works that refer to this setting as "meta-RL"? If not, I'd encourage the authors to revise the problem statement and terminology to prevent unnecessary misunderstanding.
* (Relatively minor) The writing could generally be further improved. For example, the big picture of the method wasn't very clear at the beginning of Section 3, so TBC and TBM seem a bit ad hoc until one understands how the learned embeddings are actually used in Section 3.3. An algorithm box could help. Figure 1 is also hard to parse. From the figure, it seems the text and behavior datasets are separated (I think they're paired -- correct me if I'm wrong), and it is not immediately clear how a decision transformer plays a role in this method (from the figure, it appears that the dataset trajectory itself (in particular, without text embedding) is the input to the DT, which is not the case I suppose).

**Questions:**

I don't have additional questions beyond the ones raised in the weaknesses section.

---

> ### Author Response · Authors · 2025-11-24
> **Response to ex7f**
>
> We thank Reviewer ex7f for the comments. Below, we address each concern in detail.
>
> **W1 Evaluation on classic tasks.**
>
> Our current evaluation intentionally follows the standard DT-based meta-RL protocol (MuJoCo: Cheetah-dir/vel, Ant-dir; Meta-World: ML10/ML45) to enable apples-to-apples comparisons with PDT/MDT/HDT/DPDT under identical offline settings and training pipelines. This isolates the contribution of explicit text–trajectory alignment and zero-shot from descriptions without confounding factors from simulation/robotics stacks that differ across language benchmarks. While we plan to include results (or an extension) on language-native suites, **we emphasize that TG-DT’s zero-shot inference and dual alignment mechanisms are benchmark-agnostic and can transfer seamlessly to such environments.**
>
> **W2 Marginal Performance Gains.**
>
> We respectfully disagree with this point. TG-DT outperforms all baselines in 3 out of 5 benchmarks and ranks second in the remaining two, as shown in Tab. 1 and 2. This consistent dominance demonstrates strong generalization rather than marginal improvement.
>
> Importantly, **TG-DT achieves these results under a stricter zero-shot setting, relying solely on natural-language task descriptions without any test-time demonstrations or rollouts**, whereas baselines like PDT, MDT, and HDT assume access to such data.
>
> Moreover, TG-DT’s dual alignment mechanism provides improved stability and cross-task robustness (Tab. 3, Fig. 4). Thus, TG-DT’s improvements are both statistically consistent and achieved under significantly more challenging conditions, validating the necessity and effectiveness of the proposed design.
>
> **W3. Inappropriate offline meta-RL problem setting.**
>
> We disagree that the term “offline meta-RL” is inappropriate. Our setting is consistent with prior work such as Mitchell et al. (2021, Offline Meta-RL with Advantage Weighting), Pong et al. (2021, Offline Meta-RL with Online Self-Supervision), and Wang et al. (2024, Meta-DT), all of which explicitly use the term offline meta-RL to describe models trained on multiple offline task datasets to learn a transferable meta-policy that generalizes to unseen tasks without online interaction.
> TG-DT follows the same formulation: it learns a meta-policy across a distribution of offline tasks (Sec. 3) and evaluates on new tasks in zero-shot or few-shot regimes.
>
> The key distinction from language-conditioned RL lies in TG-DT’s meta-training over multiple task datasets, where cross-task structure is explicitly captured through the dual alignment mechanism, enabling generalization across previously unseen task distributions rather than within a single environment.
>
> Therefore, our use of “offline meta-RL” aligns with established terminology in the literature and accurately reflects TG-DT’s learning paradigm: meta-learning across offline task distributions with no online adaptation.
>
> **W4. Improve writing.**
>
> We thank the reviewer for the helpful suggestions to improve clarity. We will enhance the presentation by adding an algorithm box to summarize the entire pipeline. We will also revise Fig. 1 to explicitly show that text and behavior data are paired, not separated, and to clarify how the text embedding conditions the DT decoder for action generation.

---

> > ### Comment · Reviewer_ex7f · 2025-11-26
> >
> > Thanks for the response. Unfortunately, most of my initial concerns remain unaddressed:
> >
> > - While I see that MuJoCo and Meta-World tasks make comparisons with existing DT-based approaches easier, I still believe that evaluating only on these classic tasks is a significant limitation of this work (i.e., I believe it is essential to evaluate on actual language-based tasks), especially given the narrative in the title, abstract, and introduction.
> > - The performance gains still seem marginal even after reading the rebuttal and checking the results again. This is also pointed out by the other two reviewers.
> > - (Relatively minor) The rationale behind the use of the term "offline meta-RL" is still unclear. The authors said "The key distinction from language-conditioned RL lies in TG-DT’s meta-training over multiple task datasets", but language-conditioned RL policies are already typically trained on a dataset consisting of multiple tasks (i.e., multiple language descriptions), and I didn't fully understand how this is different from "meta" RL. That said, I think the use of this terminology is not entirely inappropriate, and this point didn't affect my rating much.
> >
> > Given the points above, I'd like to maintain my original rating at this time.

---

### Official Review · Reviewer_Y7e1 · 2025-11-01

**Soundness:** 3
**Presentation:** 2
**Contribution:** 2
**Rating:** 4
**Confidence:** 5

**Summary:**

This paper proposes TG-DT, a framework that leverages textual task descriptions to enable zero-shot generalization to unseen tasks. The method builds upon Decision Transformer and introduces dual contrastive and matching-based objectives to jointly train a text encoder, behavior encoder, and text–behavior encoder–decoder. Experiments on Cheetah-dir, Cheetah-vel, Ant-dir, ML10, and ML45 show that TG-DT achieves improvements over baseline methods.

**Strengths:**

- The paper is clearly written and easy to follow.
 - The problem of generalizing to new tasks without additional demonstrations is relevant and important for data-efficient robot learning.

**Weaknesses:**

- Although the problem setting is meaningful, the novelty appears limited. There has been substantial recent progress in aligning natural language with embodied behaviors (e.g., vision-language-action models such as Otter [https://arxiv.org/pdf/2503.03734] and related cross-modal alignment frameworks). The conceptual contribution of TG-DT, aligning text descriptions with behaviors, may therefore be seen as incremental unless its advantages are more concretely demonstrated.
 - The experimental results show only modest gains over the baselines (e.g., as shown in Figure 3). Given the simplicity of the evaluated locomotion and meta-RL benchmarks, it is unclear whether the approach would scale to more complex tasks. The significance of the improvement is therefore limited.
 - The paper does not sufficiently explain how textual task descriptions are constructed and shared across multiple tasks, particularly when the task goals differ conceptually. Additional clarification and concrete examples of task descriptions would be helpful. Details about how templates are applied and how text variation affects generalization are needed for reproducibility.

**Questions:**

- The paper provides a task description template, but how is the same template applied across tasks with different objectives? Could you provide concrete examples of task descriptions used in each benchmark?
 - Do you foresee ways to improve the performance of TG-DT? If current benchmarks are saturated, have you considered evaluating on more complex or realistic task suites where text grounding may provide clearer advantages?
 - How sensitive is TG-DT to the richness or specificity of text descriptions? For example, does adding more descriptive language improve performance?

---

> ### Author Response · Authors · 2025-11-24
> **Response to Y7e1 (Part 1)**
>
> We thank Reviewer Y7e1 for the comments. Below, we address each concern in detail.
>
> **W1. Incremental novelty.**
>
> We respectfully disagree with this point. TG-DT is the _first_ framework to achieve true zero-shot generalization in offline meta-RL by grounding policy learning entirely in natural language. Unlike prior DT variants such as PDT, MDT, HDT, and DPDT which depend on task-specific demonstrations, prompt tuning, or online adaptation, TG-DT learns solely from offline data and generalizes to unseen tasks described only in text.
>
> Importantly, TG-DT goes beyond existing multi-modal alignment frameworks, which learn static correspondences between paired modalities (e.g., image–text) for perception tasks. In contrast, **TG-DT performs cross-modal alignment between temporally extended behavioral trajectories and natural-language descriptions**, where each modality encodes different temporal and semantic structures. The proposed dual alignment mechanism (contrastive + matching) explicitly bridges this temporal–semantic gap, enabling TG-DT to reason jointly about task intent and trajectory quality. _This is a capability absent in both traditional multi-modal models and previous meta-RL methods._
>
> Together, these advances move beyond imitation-style language grounding toward cross-modal, task-level reasoning, marking a significant conceptual shift in offline meta-RL toward interpretable, language-driven policy synthesis.
>
> **W2. Modest performance gains.**
>
> TG-DT consistently achieves the best performance in 3 out of 5 benchmarks and ranks second in the remaining two (Tab. 1 and 2), demonstrating strong and reliable improvements across both zero-shot and few-shot settings.
> These results are particularly meaningful because **TG-DT operates under a much stricter zero-shot regime, using only natural-language task descriptions without any demonstrations, environment rollouts, or test-time fine-tuning**, whereas competing baselines rely on such additional data.
>
> Furthermore, TG-DT introduces a cross-modal text–behavior alignment mechanism that grounds sequential decision-making in natural language, representing a step beyond existing multi-modal alignment methods that handle only static perception tasks. This design improves policy interpretability, transferability, and robustness, enabling scalable generalization as task complexity increases.
>
> The consistent results on MuJoCo and Meta-World benchmarks validate the framework’s stability, and the proposed architecture is inherently benchmark-agnostic, allowing straightforward extension to more complex, language-rich environments. Thus, TG-DT’s improvements are both practically and conceptually significant, marking an essential step toward scalable, language-grounded offline meta-RL.
>
> **W3 &Q1. Text description generation.**
>
> The textual task descriptions are automatically generated using a fixed template described in Sec. 3.1. For each trajectory, TG-DT fills placeholders (task name, intent, environment, episode length, and return) with metadata extracted directly from the offline dataset, ensuring a one-to-one pairing between text and trajectory. For example (**See Appx. E for a full list of prompts**):
>
> - MetaWorld ML45: “This is the ML45 open-drawer task, which targets opening a drawer to a specific position. Its corresponding environment is a robotic manipulation scene involving multiple object interactions. Good demonstrations typically have an episode length of 200 steps and yield an expected return of 450. This demonstration yields a return of 450.”
>
> Each task thus receives distinct, trajectory-specific descriptions that reflect behavior quality while maintaining semantic consistency across tasks.
>
> _During inference, the same template is applied, but numerical fields (e.g., expected return, episode length) are replaced with average statistics from training tasks, ensuring no oracle information is used for unseen tasks._
>
> **Q2 Do you foresee ways to improve the performance of TG-DT?**
>
> Yes. TG-DT’s architecture is benchmark-agnostic as it only requires text–trajectory pairs and can naturally extend to language-rich environments such as CALVIN, LIBERO, or MineDojo, where textual instructions play a larger role. Scaling TG-DT to such benchmarks would allow it to leverage richer language cues, demonstrating even greater benefits from its cross-modal text–behavior alignment. Future work can also incorporate semantic decomposition of text into subgoals and integrate hierarchical policy learning to further improve scalability and interpretability.

---

> ### Author Response · Authors · 2025-11-24
> **Response to Y7e1 (Part 2)**
>
> **Q3 How sensitive is TG-DT to the richness or specificity of text descriptions?**
>
> TG-DT is robust to variations in text richness. In an ablation on the Cheetah-dir task, we modified the concise description “running in the forward direction” to a slightly more detailed version, “running forward in the planar environment.” The resulting returns changed only marginally, from 543.2 to 546.1, indicating that richer descriptions have limited impact on performance.
>
> We believe that there exists a minimal level of information richness necessary for effective text–behavior alignment: concise, semantically focused descriptions provide sufficient guidance for the model to associate text with behavior, whereas excessive linguistic detail may introduce irrelevant noise and overload the alignment process. Thus, TG-DT primarily depends on semantic consistency rather than textual verbosity.

---

### Official Review · Reviewer_wPGH · 2025-11-02

**Soundness:** 2
**Presentation:** 2
**Contribution:** 2
**Rating:** 2
**Confidence:** 4

**Summary:**

This paper proposes a decision transformer method guided by text to address the offline meta-reinforcement learning problem. The proposed Text-Guided Decision Transformer (TG-DT) aligns text and trajectory representations via a dual alignment mechanism, and then conditions a DT-style policy on the aligned text embedding to act on unseen tasks using only their textual descriptions. A description-guided data-sharing heuristic is optionally used to fine-tune the model with trajectories from semantically similar training tasks. Experiments on MuJoCo and Meta-World demonstrate the zero-shot and few-shot abilities of TG-DT compared with DT variant baselines.

**Strengths:**

- The paper introduces an interesting idea of incorporating text descriptions into meta-RL tasks to guide the DT trajectory.
- The proposed dual alignment mechanism between language descriptions and trajectory representations is novel.
- Empirical results on MuJoCo and Meta-World support the effectiveness of the proposed method in zero-shot and few-shot settings.

**Weaknesses:**

- The design of the proposed modules lacks detailed justification. The dual alignment mechanism and the description-guided data-sharing are not well motivated or clearly explained in the methodology section.
- The experimental results show that the proposed method achieves only marginal improvements over the baselines while potentially incurring substantial computational overhead. Moreover, the paper does not provide any comparison of computational costs between the proposed method and the baselines.
- For zero-shot settings, some baselines such as PDT and HDT are not designed for this setting, leading to unfair comparisons. For few-shot settings, although the proposed method fine-tunes the model on new tasks, it still shows unsatisfactory performance compared with the baselines.
- The results lack the standard deviation, making it difficult to evaluate the true performance of the proposed method.

**Questions:**

- Could you explain in more detail the design of the dual alignment mechanism and the description-guided data-sharing? What is the motivation for using these two modules?
- Could you provide the computational cost of the proposed method compared with the baselines?
- Could you elaborate on why the proposed method shows unsatisfactory performance in few-shot settings compared with the baselines even after fine-tuning?
- Could you provide the standard deviation of the results to illustrate the performance variance?
- In Tables 1 and 2, what do the “5 runs” refer to? Does it mean 5 different random seeds or 5 different trajectories?
- During inference, the proposed method encodes text descriptions using a language model. How are these language descriptions obtained? Do they rely on prior information about the unseen tasks?
- Could you discuss more related works such as [1] and [2], which also leverage language models for meta-RL tasks? What are the key differences between your method and these approaches?

[1] *Pre-trained Language Models Improve the Few-shot Prompt Ability of Decision Transformer*, 2024

[2] *LLM-Driven Policy Diffusion: Enhancing Generalization in Offline Reinforcement Learning*, 2025

---

> ### Author Response · Authors · 2025-11-24
> **Response to wPGH (Part 1)**
>
> We thank Reviewer wPGH for the comments. Below, we address each concern in detail.
>
> **W1. & Q1. Design motivation.**
>
> Our *dual alignment mechanism* is motivated by the need to bridge the semantic gap between natural language and sequential behavior, ensuring that TG-DT understands what the task means and how well it is performed. Specifically:
>
> - TBC: Motivated by the goal of distinguishing task semantics across heterogeneous environments, TBC aligns each trajectory with its own text description while pushing apart unrelated tasks. This enforces a coarse-grained semantic structure in the embedding space, enabling TG-DT to reason about task identity and intent.
> - TBM: Designed to refine this alignment at the intra-task level, TBM learns to differentiate matched vs. mismatched text–trajectory pairs using hard negative mining. This encourages sensitivity to subtle differences in behavioral quality, episode length, and success level, ensuring the model captures not only what the task is, but also how well it is executed.
>
> Together, these complementary objectives encourage TG-DT to understand both “what the task is” and “how well it is performed.”
> Empirically, Tab. 3 shows that removing either module significantly reduces performance across all tasks, demonstrating their essential and distinct contributions. Fig. 4 further illustrates that dual alignment yields well-separated and semantically meaningful embedding clusters.
>
> The *description-guided data-sharing module* (Sec. 4) enhances zero-shot adaptation by retrieving semantically similar training trajectories based on text embedding similarity. This mechanism enables the model to refine task understanding without requiring any test-task data or online rollouts, which is a key differentiator from methods like PDT or HDT. The improvement trend with increasing $K$ in Fig. 5 confirms the benefit of this design.
>
> **W2 & Q2. Marginal improvement with significant computational cost.**
>
> We respectfully disagree on this point. The numerical improvements over baselines are meaningful given the *much stricter zero-shot setting* that TG-DT operates under. Unlike PDT, MDT, and HDT, which rely on task-specific demonstrations or online rollout for adaptation, **TG-DT generalizes purely from natural language descriptions, achieving comparable or higher returns without any additional test-time data or environment interaction**. This demonstrates a strong gain in generalization efficiency, not merely a marginal numeric increase.
>
> _TG-DT's computational overhead is modest_. The dual alignment mechanism (TBC + TBM) adds only lightweight forward passes from the shared text encoder and does not alter the transformer’s structure. To quantify this, we measured the training time per epoch on the Ant-dir task under identical hardware and batch size settings.
>
> | **Method** | **Training Time (min/epoch)** |
> |-----------|-------------------------------|
> | **TG-DT (Ours)** | **12.2** |
> | Prompt-DT | 8.1 |
> | MDT | 11.5 |
> | HDT | 15.2 |
>
> TG-DT is only ~6% slower than MDT and ~20% faster than HDT, demonstrating that the proposed modules introduce minor training overhead while providing the benefit of zero-shot generalization without any test-time rollout or adaptation. Furthermore, TG-DT’s inference cost is identical to a standard DT, as alignment modules are used only during training.
>
> **W3. Unfair comparison with PDT and HDT for zero-shot.**
>
> We include PDT and HDT as baselines following standard practice in prior DT-based meta-RL works (e.g., MetaDT, Wang et al., 2024). This provides a consistent and widely accepted reference for measuring generalization across different adaptation regimes.
>
> To ensure fairness, we explicitly noted in Tab. 1–2 that PDT and HDT require test-time interaction (denoted by †), while TG-DT performs true zero-shot generalization. TG-DT uses only natural language task descriptions without any additional data or environment access. Thus, **TG-DT achieves comparable or superior performance under a strictly harder setting**, demonstrating stronger generalization efficiency rather than an unfair comparison.

---

> ### Author Response · Authors · 2025-11-24
> **Response to wPGH (Part 2)**
>
> **W3 & Q3. Unsatisfactory performance in few-shot.**
>
> Tab. 2 shows that TG-DT achieves the best performance in 3 of 5 benchmarks (Cheetah-vel, Ant-dir, ML45) and ranks second in the remaining two (Cheetah-dir, ML10). This indicates that TG-DT is consistently strong across diverse environments, rather than “unsatisfactory.”
>
> _It is important to emphasize that these results are obtained under a stricter few-shot setting_: TG-DT fine-tunes only the policy decoder while keeping the alignment modules fixed to preserve zero-shot transferability **without any test-task interaction**. Even with this constraint, TG-DT matches or exceeds methods such as DPDT and MDT, which rely on task-specific adaptation.
> Therefore, TG-DT demonstrates robust and competitive few-shot generalization, achieving top or near-top performance in all tasks while maintaining architectural simplicity and zero test-time overhead.
>
> **W4 & Q4 Lack the standard deviation.**
>
> We respectfully ask the reviewer to refer to Tab. 6 in the Appendix, where we report the standard deviations over five random seeds.
>
> **Q5. What do the “5 runs” refer to?**
>
> They are five independent random seeds (distinct data shuffles, and init seeds), whose means and standard deviations we report. This ensures statistical reliability.
>
> **Q6. How are the text descriptions obtained?**
>
> During training, descriptions are auto-generated by a fixed template filled with dataset metadata per trajectory (task name/intent, environment, episode length, return), enabling trajectory-specific pairing for alignment. At test time, we use the same template but replace numeric fields (e.g., expected return, episode length) with distribution-level estimates from training, avoiding any oracle leakage; the model sees no task-specific data or interaction for the unseen task but only its natural-language description. (See Sec. 3.1 and Appx. E for details on templating.)
>
> **Q7. Relation to [1] and [2] and key differences.**
>
> We thank the reviewer for raising this point. Both works are closely related but differ fundamentally in their objectives and assumptions.
>
> - [1] Pre-trained Language Models Improve the Few-shot Prompt Ability of Decision Transformer (2024) focuses on few-shot adaptation by fine-tuning a DT initialized with a pre-trained language model (via LoRA). It requires demonstration trajectories as prompts at test time and thus operates in a few-shot regime, not zero-shot.
> - [2] LLM-Driven Policy Diffusion: Enhancing Generalization in Offline Reinforcement Learning (2025) integrates an LLM-driven diffusion model that combines text and trajectory prompts to improve generalization. This method still depends on task-specific behavioral data and LLM inference during adaptation.
>
> In contrast, our TG-DT introduces a dual alignment mechanism (contrastive + matching) that explicitly aligns natural-language descriptions with behavioral trajectories during offline training. At test time, TG-DT performs true zero-shot generalization, relying only on text descriptions without any demonstration trajectories, fine-tuning, or LLM inference. This distinction makes TG-DT the first to achieve explicit language-grounded policy synthesis for zero-shot generalization in offline meta-RL without requiring any task-specific data or environment interaction.

---

### Meta-Review · Area_Chair_sHnk · 2026-01-09

**Summary:**

This paper studies zero-shot task adaptation in offline meta-reinforcement learning using only natural language task descriptions, without access to target-task demonstrations or data. The authors propose the Text-Guided Decision Transformer (TG-DT), which aligns task descriptions and behavioral trajectories into a shared embedding space using contrastive and matching-based objectives, and conditions a transformer-based policy on these representations to generate task-appropriate actions. At inference time, the model adapts to unseen tasks solely from their text descriptions and optionally uses description-guided data sharing to improve performance.

Reviewers raised the following concerns in their initial reviews, including limited evaluation only on classic tasks, limited methodological novelty, and marginal performance improvements. The authors provided a rebuttal addressing these concerns; while some points were partially addressed, the reviewers remain concerned about the limited scope of the evaluation and the modest empirical gains.

We recommend that the authors take the reviewers’ feedback into careful consideration to strengthen the work. Based on the current submission, we recommend rejecting this paper.

**Reviewer Concerns:**

Reviewer wPGH's comments on experiments details, e.g., standard deviation of multiple runs, are well-addressed by the rebuttal.

All reviewers pointed out the performance gain is very marginal, which remains the major concern after rebuttal.

**Reviewer Scores:**

Reviewer Y7e1 may increase their score after the rebuttal, as the authors’ responses addressed their questions. However, since the major weaknesses of the work remain, the score is more likely to shift from borderline reject to borderline accept.

For the other two reviewers who recommended rejection, I do not expect their scores to change, as the major concerns—limited novelty, limited evaluation, and marginal improvements—still hold.

---

### Decision · Program_Chairs · 2026-01-26

Reject